# Youth's Social Environments: Associations with Mental Problems and Achievement of Developmental Milestones in Times of Crises

Leanne A. C. van Est-Bitincka [1,2,*], Hilde D. Schuiringa [2], Paul T. van der Heijden [1,3], Marcel A. G. van Aken [2] and Odilia M. Laceulle [2]

1   Reinier van Arkel, 5211 LJ 's-Hertogenbosch, The Netherlands
2   Department of Developmental Psychology, Utrecht University, 3584 CS Utrecht, The Netherlands
3   Behavioural Science Institute, Radboud University Nijmegen, 6525 GD Nijmegen, The Netherlands
*   Correspondence: l.van.est@reiniervanarkel.nl

**Abstract:** So far, many studies indicated that youth experience mental problems during crises, such as the COVID-19 crisis, but little attention has been paid to the relation to age-adequate functioning and its association to layered social environments. This study addresses this gap by investigating the association between social environments (i.e., household, friends, and neighbourhood) during the COVID-19 crisis with youth's mental problems and age-adequate functioning. In total, 673 youth (mean age = 19.87, 73.4% girls) were surveyed online during the COVID-19 outbreak. In line with predictions, worse contact with household members was associated with more internalizing symptoms. A lack of privacy was associated with more internalizing and externalizing symptoms and difficulties achieving personal and school and professional milestones. Living with a vulnerable other was associated with more internalizing symptoms and difficulties achieving school and professional milestones. Worse contact with friends was associated with difficulty achieving social milestones. Additionally, neighbourhood risk moderated the association between living with a vulnerable other and school and professional milestones. A lack of privacy stood out as the most important factor associated to youth's mental problems and achievement of developmental milestones. Future research should indicate to what extent these findings are COVID-19 crisis-specific or can generalize to other crises.

**Keywords:** youth; crisis; mental problems; developmental milestones; social environments

## 1. Introduction

Humanitarian crises, a single event or a series of events that pose a critical threat to the health, safety, or well-being of a community or large group of people [1], can have a long-lasting impact depending on the context people are in. The COVID-19 crisis has recently passed, wars are going on, and we anticipate future crises [2]. Adolescents and young adults, hereafter referred to as youth, may be particularly impacted by such crises. They are in a developmental period characterized by large psychological and social role changes [3]. In general, youth are found to experience a variety of mental problems during crises, such as depression and anxiety [4]. Most studies to date, however, have not simultaneously considered the relation to youth's age-adequate functioning, such as becoming more independent, nor the associations between youth's functioning and layered social environments. Therefore, the current research will focus on youth's mental problems and age-adequate functioning and study how they are associated with youth's layered social environments of their household, friends, and neighbourhood.

An important, proximal social environment for youth is their household. Studies indicate that better quality relationships between youth and their family are protective against youth's internalizing and externalizing symptoms in different crises [5–10]. For

other household members, such as roommates, this has also been shown but not yet in times of crises [11,12]. While good relationships with household members can provide protection during crises, these relationships can also take a toll on youth. It has been shown that having a vulnerable household member, for instance having a family member with a mental illness or underlying illness during the COVID-19 crisis or HIV pandemic, was related to youth's internalizing and externalizing symptoms [13–15]. In addition to fostering positive relationships with household members, it is also important for youth to experience privacy to achieve independence from them [16,17]. A study theorised that quarantine could lead to decreased privacy in the home and, consequently, higher stress for children and adolescents [18]. To our knowledge, this has not been empirically studied yet, but is important to study as a lack of privacy could potentially be unfavourable for youth.

Not only is the household important, but friends also become especially important as a social environment for youth. In line with results on household members, better quality relationships with friends were also protective against internalizing symptoms in youth exposed to different crises [5,6,8,10]. On the association with externalizing symptoms in times of crises, no previous research has been performed.

Youth function not only in the social environments of the household and friends but also in the distal social environment of their neighbourhood, which grows in importance when children grow up and become adolescents [19,20]. It has been documented that neighbourhoods are stratified by place and vary by economic and social (in)equality [21]. Youth living in disadvantaged neighbourhoods face greater risks of internalizing and externalizing symptoms [22]. Furthermore, in the most disadvantaged neighbourhoods, the association between household factors and youth's mental problems were strongest [23], indicating a moderating effect of disadvantaged neighbourhoods on social relationships and mental problems. These studies investigated neighbourhoods on multiple dimensions [22,23], thereby making it possible to study multiple risk effects. This is important because risk in neighbourhoods can be recognised and measured on distinct dimensions, such as levels of poverty and unemployment and lack of resources (e.g., [24]). Concentrated neighbourhood risk is likely to have even more important implications on youth's functioning, as exposure to positive role models decreases and the number of risk factors increases [21]. So far, neighbourhood risk as a factor with multiple dimensions in times of crises has not been studied, also studies on its association to youth mental problems have been lacking. A few studies provide preliminary results. For example, communities with lower pre-existing socioeconomic structures have shown a lower recovery rate from disasters [25] and living in urban areas, compared to rural regions, was protective against internalizing symptoms during the COVID-19 crisis [26,27]. Additionally, studies on more individual or family levels, that could possibly extend to the neighbourhood level, showed that low education or income was related to mental problems in children and youth during a crisis [26,28–30]. Although these studies are informative, a more comprehensive view of neighbourhood risk during crises is needed to understand the association between neighbourhood risk and youth's mental problems. That is, to understand neighbourhood risk, multiple neighbourhood dimensions are needed for greater coverage of the construct and to understand concentration of risk factors [21,31]. Dimensions on the neighbourhood level to include are the level of income, employment, education, ethnicity, health, crime, obstacles to housing and services, and living environment [24,32].

So far, many gaps in the literature still exist on associations between youth's social environments and their mental problems during crises. Unfortunately, even less is known on youth's age-adequate functioning during crises. Age-adequate functioning can be assessed by focusing on youth's achievement of developmental milestones [33]. Developmental milestones are critical tasks that characterize specific stages of life. Examples of youth-specific developmental milestones are becoming more independent, forming an identity, maintaining relationships with parents and friends, finishing an education, and starting a job [33,34]. These milestones can be roughly divided into personal, school and professional, and social milestones. The extent to which youth successfully achieve these milestones

foretells their personal and social adjustment as adults, as is argued by developmental theorists [19,35,36] and demonstrated in empirical work. For instance, failing to graduate from high school resulted in substantially lower earnings over the life course [37], both examples are of school and professional milestones. Thus, considering the potential long-term consequences of achieving developmental milestones, it is important to study youth's achievement of milestones during times of crises.

Focusing on the association between youth's household and their achievement of developmental milestones, it is known that during the COVID-19 crisis, lower levels of parental support and involvement were related to lower academic motivation and school bonding [38,39], which are examples of school and professional milestones. Another study, not in times of crises, demonstrated that a better relationship with parents was a significant predictor of achieving personal and social milestones [40]. Until now, no studies have been performed regarding relationship quality with household members and personal or social milestones during times of crises. Regarding the association between living with a vulnerable other and the achievement of milestones, it was found that children had poorer school performance when their parent had a mental illness (see [41]). However, this study only focused on school and professional milestones and was not taking place during a crisis. Additionally, no previous research focused on the association between experienced privacy in the household and achieving milestones in times of crises, although from other research it is known that youth's need for privacy is related to their need and personal milestone of becoming psychologically independent and autonomous [16,42].

No previous studies have explored the association between relationship quality with friends and youths' achievement of developmental milestones during crises. However, studies conducted in non-crisis periods have shown that students with better quality friendships have better school results and greater persistence in their studies [43], which are examples of school and professional milestones. Additionally, better relationships with friends was a significant predictor of achieving personal and social milestones, such as identity formation and experiencing trust and good communication with peers [40].

Focusing on the neighbourhood and its association to achievement of milestones, again, no previous studies have been performed examining this association in times of crises. Pointing towards a possible association are studies that demonstrated that youth living in disadvantaged neighbourhoods are at greater risk for worse achievement of school and professional milestones [32,44–46]. Additionally, pointing towards a moderating effect of neighbourhood risk, is a study that showed that disadvantaged neighbourhood friend influences were strong predictors of problems in achieving school and professional milestones [47].

In the present study, how youth are functioning during a crisis was examined, considering youth's layered social environments of the household, friends, and neighbourhood. Specifically, we looked at the associations between perceived contact with household members, living with a vulnerable other, experienced lack of privacy, perceived contact with friends, and neighbourhood risk on the one hand and youth's mental problems and achievement of developmental milestones on the other hand. In doing so, we also examined the neighbourhood risk as a putative moderating factor. As such, we built on and extended previous work by examining youth's mental problems and age-adequate functioning to obtain a better understanding of how youth are functioning during a crisis. Moreover, we used multi-dimensional data (e.g., income, crime, and health) from the national statistical office to index neighbourhood risk. Findings could inform about social environmental influences, which, in turn, may shed light on policies to alleviate negative consequences of current and future crises. We hypothesized that youth who perceived their contact with household members as worse, lived with a vulnerable other, experienced a lack of privacy, perceived their contact with friends as worse, and lived in a high-risk neighbourhood would have more mental problems and difficulties achieving developmental milestones. We also hypothesized that the links regarding household and friend factors would be especially pronounced among youth living in high-risk neighbourhoods.

## 2. Materials and Methods

### 2.1. Participants and Procedure

A total of 974 participants aged 16 to 24 years ($M$ = 19.62, $SD$ = 2.45) were asked to complete a 15 min survey online. Since only youth with complete information on all variables of interest were considered for the analyses, the sample size was reduced to $n$ = 673 youth. Their average age was 19.87 years ($SD$ = 2.40). The majority of participants identified as a girl (73.4%), were born in the Netherlands (96.4%), and lived with their parents (71.6%). There were no exclusion criteria other than age.

The study was approved by the Ethics Review Board of the Faculty of Social Sciences of Utrecht University (filed under number 20-413); participants provided their informed consent. The study took place from October 2020 to April 2021. To ensure a substantially diverse sample, we recruited participants using a three-pronged approach via (1) a school for Intermediate Vocational Education, (2) a regional public health service, and (3) a university. We used slightly different procedures for each subsample. (1) Youth from the school for Intermediate Vocational Education received an information letter via e-mail from the school and a reminder 1 week later. They could win 1 out of 200 gift vouchers worth EUR 25. (2) Youth from the regional public health service received an information letter via e-mail from the health service and a reminder 1 week later. (3) Youth from Utrecht University were informed about the study on a platform with ongoing studies that university students could take part in. They were rewarded EUR 4 or study credits with a similar value. See Figure 1 for a flow chart of participant recruitment and analyses.

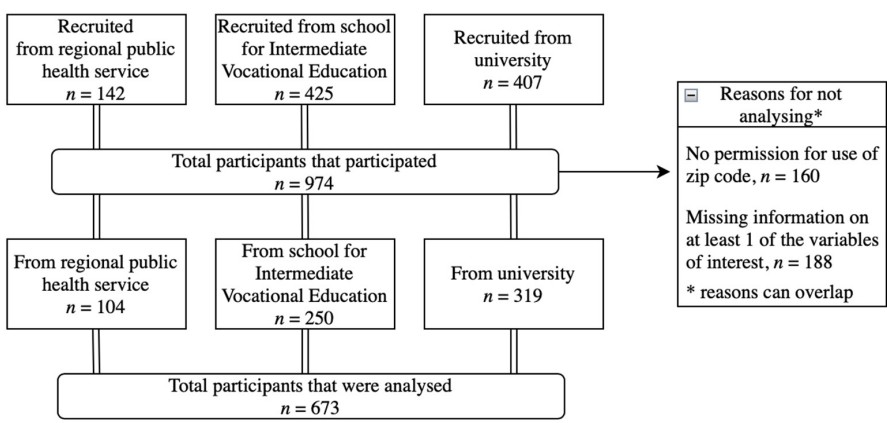

**Figure 1.** Participant flowchart.

### 2.2. Measures

Demographic information included questions about participants' age, gender, zip code (to ensure anonymity, zip codes of participants were saved in a different file, and analyses were performed on the neighbourhood risk index instead of the zip code), ethnic background, and education level. They were asked who their household members were (i.e., parents, siblings, student roommates, and other).

Household. To assess perceived social contact with household members, participants were asked with a single item to rate along a 3-point scale (i.e., worse, similar, better) how they perceived their social contact with their household members during the first lockdown compared to before. Additionally, youth were asked with a single item if they lived with someone who is vulnerable (e.g., for mental or physical reasons) to the COVID-19 virus (answer options: yes, no), and about their privacy. To measure their experiences of privacy, participants answered the statement 'During the lockdown I did not have privacy anymore (space and time for myself)' along a 3-point scale (i.e., do not agree with, agree with a little, agree with). Variables were recoded if necessary and analysed in such a way that a higher score meant worse contact, living with a vulnerable other, and a lack of privacy. The items on living with a vulnerable other and a lack of privacy were asked upon the advice of a youth panel. The youth had formulated the research question: 'What is the impact of

COVID-19 on vulnerable youth'. Then, the research team together with the youth tried to answer this question. We discussed, for example, different ways in which youth could be vulnerable from the youth's perspective and developed questionnaire items based on this.

Friends. To assess perceived social contact with friends, participants were asked with a single item to rate along a 3-point scale (i.e., worse, similar, better) how they perceived their social contact with friends during the first lockdown compared to before. The variable was analysed in such a way that a higher score meant worse contact.

Neighbourhood risk index. See Supplementary Materials for additional, detailed information. The neighbourhood risk index is a score that indicates the level of risk of a neighbourhood relative to other neighbourhoods in the Netherlands. The index is linked to zip codes. Objective, statistical data on the neighbourhoods' participants living in them were obtained via the national statistical office, Statistics Netherlands (CBS; [48]). The model of multiple risk, which underpins the index, is based on the idea of distinct domains of risk, which can be recognised and measured separately [21,24,31]. The 10 risk domains, based on the English Indices of Deprivation 2019 [24] and on the work of [32], that were included are income, employment, education, health, crime, housing, obstacles to services, green space, address density, and ethnicity. Domains were constructed separately from single or multiple component indicator(s). Following the work from [49], domain scores were then standardized. Next, the 10 standardized domain scores (0 = no risk, 1 = at risk) were added to create a total index score (range: 0–10). The cumulative risk index was categorized as a low (0 risk factors), moderate (1–2 risk factors), high (3–4 risk factors), or extremely high (>4 risk factors) risk. For the current study, due to the small number of participants with an extremely high risk, the high and extremely high risks were combined in the index as a (extremely) high risk (range: 1 = low, 2 = moderate, 3 = (extremely) high risk).

Mental problems. To assess mental problems, both internalizing and externalizing symptoms, we administered the Dutch translation of the self-reported Strengths and Difficulties Questionnaire for those aged 18 and over (Dutch SDQ s18+; [50,51]). The SDQ is rated on a 3-point Likert scale ranging from 0 (not true) to 2 (certainly true). For the current study, the internalizing and externalizing symptom scores were used. Both scales on internalizing and externalizing symptoms consist of 10 items. Scores were calculated by taking the sum of the subscales of emotional symptoms and peer relationship problems for internalizing symptoms and the sum of the subscales of conduct problems and hyperactivity/inattention for externalizing symptoms (range: 0–20). Higher scores reflect more difficulties. The SDQ has been found to be reliable and valid [52,53]. In the current sample, the Cronbach's alpha for internalizing symptoms was 0.72, and 0.67 for externalizing symptoms, which are comparable to what was found in previous research [53].

Developmental milestones. The achievement of developmental milestones was assessed using the Developmental Milestones List (DML; [54]). The DML consists of 21 items, which are reflective of youth-specific developmental milestones. The specific milestones are divided into three broader domains based on previous work on youth-specific milestones [20,55]: personal (e.g., 'To what extent are you able to become independent?'), school and professional (e.g., 'To what extent are you able to go to school/work?'), and social (e.g., 'To what extent are you able to experience it as pleasant and important to make and have friends?'), each with 7 items. All items were rated on a 7-point Likert scale ranging from −3 (not at all) to +3 (very good). Composite scores for the three scales were calculated by taking the sum of the corresponding items. Higher scores represent fewer difficulties in achieving developmental tasks. In the current sample, the Cronbach's alpha for personal milestones was 0.68, 0.81 for school and professional milestones, and 0.74 for social milestones.

### 2.3. Statistical Analyses

Prior to performing the main analyses, bivariate correlation analyses were conducted with all study variables for descriptive purposes. To analyse the association of youth's social environments during a crisis with youth's functioning, two groups of multivariate regression analyses were performed in SPSS (version 28, [56]). In the first analysis, de-

pendent variables were internalizing and externalizing symptoms. In the second analysis, dependent variables were personal, school and professional, and social milestones. Independent variables were contact with household members, living with a vulnerable other, a lack of privacy, contact with friends, and neighbourhood risk. Independent variables were centred before being used in the statistical analysis and entered simultaneously in the model. Additionally, we studied the moderation effect of neighbourhood risk on contact with household members, living with a vulnerable other, a lack of privacy, and contact with friends. All analyses were corrected for by age, education level, and gender. A conservative $p$ value of 0.01 was used to examine statistically significant findings.

## 3. Results

### 3.1. Preliminary Results

An independent samples *t*-test was performed to assess possible differences between completers ($n = 673$) and non-completers ($n = 301$). The groups were similar on most variables (all $p$'s > 0.103). However, completers had higher education ($t(968) = 5.79$, $p = 0.006$, Cohen's d = 1.43), had a better relation with their household members ($t(946) = -0.81$, $p < 0.001$, Cohen's d = 0.59), lived less often with a vulnerable other ($t(964) = -1.09$, $p = 0.042$, Cohen's d = 0.48), and achieved personal milestones with fewer difficulties ($t(892) = 0.65$, $p = 0.037$, Cohen's d = 5.81).

Descriptive statistics of the sample that were included in the following analyses are reported in Table 1 and correlation coefficients are displayed in Table 2. As shown in Table 2, internalizing symptoms show positive relations with contact with household members, living with a vulnerable other, and a lack of privacy. Externalizing symptoms show a positive relation with a lack of privacy. Personal and school and professional milestones show negative relations with contact with household members, living with a vulnerable other, and a lack of privacy. Social milestones show negative relations with a lack of privacy and contact with friends. All significant correlations between the predictor and outcome show a small effect size [57]. Furthermore, between the predictor variables, a lack of privacy shows a positive relation with worse contact with household members. Neighbourhood risk shows negative relations with worse contact with household members and living with a vulnerable other and a positive relation with worse contact with friends. These correlations show a small effect size [57]. Between the outcome variables, externalizing symptoms show a positive relation with internalizing symptoms. The milestones show negative relations with mental problems and the milestones show positive relations with the other milestones. These correlations show medium to large effect sizes [57].

**Table 1.** Descriptive Statistics.

| Variable | *M*/% | *SD* | Min | Max |
|---|---|---|---|---|
| Age | 19.87 | 2.40 | 16 | 24 |
| Education level | 12.86 | 1.43 | 4 | 14 |
| Gender (boy) | 26.6% | - | - | - |
| Contact with household members | 2.18 | 0.62 | 1 | 3 |
| Lack of privacy | 1.53 | 0.63 | 1 | 3 |
| Living with a vulnerable other | 36.3% | - | - | - |
| Contact with friends | 1.55 | 0.69 | 1 | 3 |
| Neighbourhood risk index | 1.57 | 0.64 | 1 | 3 |
| Low risk | 51.0% | | | |
| Moderate risk | 40.9% | | | |
| (Extremely) high risk | 8.2% | | | |
| Internalizing symptoms | 5.54 | 3.36 | 0 | 17 |
| Externalizing symptoms | 5.32 | 2.94 | 0 | 16 |
| Personal milestones | 9.28 | 5.62 | −16 | 21 |
| School and professional milestones | 9.53 | 5.84 | −13 | 21 |
| Social milestones | 11.48 | 5.00 | −8 | 21 |

**Table 2.** Pearson's Correlations for Study Variables.

| Variable | 1 | 2 | 3 | 4 | 5 | 6 | 7 | 8 | 9 | 10 | 11 | 12 | 13 |
|---|---|---|---|---|---|---|---|---|---|---|---|---|---|
| 1 Age | 1 | | | | | | | | | | | | |
| 2 Education level | 0.60 * | 1 | | | | | | | | | | | |
| 3 Gender (boy) | −0.10 | −0.16 * | 1 | | | | | | | | | | |
| 4 Contact with household members | −0.10 * | −0.09 | −0.02 | 1 | | | | | | | | | |
| 5 Vulnerable other | −0.24 * | −0.27 * | 0.00 | 0.05 | 1 | | | | | | | | |
| 6 Lack of privacy | −0.03 | 0.01 | −0.13 * | 0.22 * | 0.06 | 1 | | | | | | | |
| 7 Contact with friends | 0.09 | 0.10 | 0.01 | −0.01 | −0.03 | 0.09 | 1 | | | | | | |
| 8 Neighbourhood risk | 0.28 * | 0.23 * | −0.03 | −0.13 * | −0.14 * | −0.06 | 0.14 * | 1 | | | | | |
| 9 Internalizing problems | −0.06 | −0.04 | −0.25 * | 0.16 * | 0.14 * | 0.22 * | 0.07 | −0.01 | 1 | | | | |
| 10 Externalizing problems | −0.16 * | −0.19 * | 0.14 * | 0.04 | 0.08 | 0.12 * | −0.04 | −0.05 | 0.30 * | 1 | | | |
| 11 Personal milestones | 0.14 * | 0.07 | 0.04 | −0.14 * | −0.11 * | −0.22 * | −0.07 | 0.03 | −0.44 * | −0.33 * | 1 | | |
| 12 School and professional milestones | 0.08 | 0.06 | −0.08 | −0.12 * | −0.13 * | −0.14 * | −0.03 | 0.04 | −0.35 * | −0.50 * | 0.55 * | 1 | |
| 13 Social milestones | 0.00 | 0.03 | −0.03 | −0.09 | −0.07 | −0.12 * | −0.11 * | 0.03 | −0.41 * | −0.27 * | 0.48 * | 0.49 * | 1 |

* $p < 0.01$ (two-tailed).

### 3.2. Main Analyses

### 3.2.1. Mental Problems

Results from the first multiple regression analyses showed that youth who had a worse contact with household members, youth who lived with a vulnerable other, and youth who experienced a lack of privacy reported more internalizing symptoms. Youth who experienced a lack of privacy also reported more externalizing symptoms. All effect sizes were small. Contact with friends, neighbourhood risk, and the moderation effect of neighbourhood risk were not associated to mental problems. In conclusion, the hypotheses on internalizing symptoms were supported for contact with household members, living with a vulnerable other, and a lack of privacy. The hypotheses on contact with friends, neighbourhood risk, and the moderating effect of neighbourhood risk were not supported. The hypothesis on externalizing symptoms was supported for a lack of privacy, but not for all the other factors. See Table 3 for parameter estimates.

**Table 3.** Results of Multiple Regression Analysis.

| Dependent Variable | Parameter | B | SE | *p* | 95% CI | | η² |
|---|---|---|---|---|---|---|---|
| | | | | | LL | UL | |
| Internalizing problems | Contact with household members | 0.62 | 0.21 | 0.003 | −1.026 | −2.17 | 0.014 |
| | Vulnerable other | 0.79 | 0.27 | 0.003 | −1.316 | −0.267 | 0.013 |
| | Lack of privacy | 0.85 | 0.20 | <0.001 | 0.451 | 1.245 | 0.026 |
| | Contact with friends | 0.30 | 0.18 | 0.097 | −0.652 | 0.055 | 0.004 |
| | Neighbourhood risk | 0.61 | 0.43 | 0.156 | −0.233 | 1.446 | 0.003 |
| | Risk × Household members | −0.15 | 0.32 | 0.635 | −0.483 | 0.791 | 0.000 |
| | Risk × Vulnerable other | 0.45 | 0.42 | 0.290 | −1.271 | 0.380 | 0.002 |
| | Risk × Lack of privacy | −0.30 | 0.33 | 0.356 | −0.949 | 0.341 | 0.001 |
| | Risk × Friends | −0.26 | 0.29 | 0.365 | −0.305 | 0.827 | 0.001 |
| Externalizing problems | Contact with household members | −0.02 | 0.19 | 0.912 | −0.346 | 0.387 | 0.000 |
| | Vulnerable other | 0.19 | 0.24 | 0.425 | −0.669 | 0.282 | 0.001 |
| | Lack of privacy | 0.64 | 0.18 | <0.001 | 0.283 | 1.003 | 0.018 |
| | Contact with friends | −0.12 | 0.16 | 0.462 | −0.200 | 0.440 | 0.001 |
| | Neighbourhood risk | −0.07 | 0.39 | 0.859 | −0.829 | 0.692 | 0.000 |
| | Risk × Household members | −0.30 | 0.29 | 0.309 | −0.278 | 0.876 | 0.002 |
| | Risk × Vulnerable other | 0.32 | 0.38 | 0.399 | −1.070 | 0.427 | 0.001 |
| | Risk × Lack of privacy | 0.09 | 0.30 | 0.758 | −0.493 | 0.676 | 0.000 |
| | Risk × Friends | 0.30 | 0.26 | 0.245 | −0.817 | 0.209 | 0.002 |
| Personal milestones | Contact with household members | −0.72 | 0.36 | 0.042 | 0.025 | 1.418 | 0.006 |
| | Vulnerable other | −0.88 | 0.46 | 0.056 | −0.024 | 1.783 | 0.006 |
| | Lack of privacy | −1.71 | 0.35 | <0.001 | −2.397 | −1.029 | 0.035 |
| | Contact with friends | −0.52 | 0.31 | 0.091 | −0.084 | 1.132 | 0.004 |
| | Neighbourhood risk | −0.81 | 0.74 | 0.271 | −2.256 | 0.634 | 0.002 |
| | Risk × Household members | 0.15 | 0.56 | 0.787 | −1.247 | 0.945 | 0.000 |
| | Risk × Vulnerable other | −1.00 | 0.72 | 0.169 | −0.425 | 2.417 | 0.003 |
| | Risk × Lack of privacy | −0.05 | 0.57 | 0.926 | −1.163 | 1.058 | 0.000 |
| | Risk × Friends | 0.01 | 0.50 | 0.989 | −0.981 | 0.968 | 0.000 |

**Table 3.** *Cont.*

| Dependent Variable | Parameter | B | SE | *p* | 95% CI | | η² |
|---|---|---|---|---|---|---|---|
| | | | | | LL | UL | |
| School and professional milestones | Contact with household members | −0.65 | 0.37 | 0.080 | −0.078 | 1.378 | 0.005 |
| | Vulnerable other | −1.44 | 0.48 | 0.003 | 0.495 | 2.384 | 0.013 |
| | Lack of privacy | −1.18 | 0.36 | 0.001 | −1.898 | −0.469 | 0.016 |
| | Contact with friends | −0.26 | 0.32 | 0.425 | −0.377 | 0.894 | 0.001 |
| | Neighbourhood risk | −1.07 | 0.77 | 0.167 | −2.575 | 0.446 | 0.003 |
| | Risk × Household members | 0.28 | 0.58 | 0.635 | −1.423 | 0.868 | 0.000 |
| | Risk × Vulnerable other | −2.77 | 0.76 | <0.001 | 1.284 | 4.255 | 0.020 |
| | Risk × Lack of privacy | 0.94 | 0.59 | 0.113 | −0.223 | 2.099 | 0.004 |
| | Risk × Friends | −0.75 | 0.52 | 0.150 | −0.271 | 1.766 | 0.003 |
| Social milestones | Contact with household members | −0.51 | 0.32 | 0.114 | −0.124 | 1.147 | 0.004 |
| | Vulnerable other | −0.62 | 0.42 | 0.141 | −0.206 | 1.442 | 0.003 |
| | Lack of privacy | −0.75 | 0.32 | 0.019 | −1.369 | −0.122 | 0.008 |
| | Contact with friends | −0.73 | 0.28 | 0.010 | 0.172 | 1.281 | 0.010 |
| | Neighbourhood risk | −0.77 | 0.67 | 0.252 | −2.087 | 0.540 | 0.002 |
| | Risk × Household members | −0.06 | 0.51 | 0.905 | −0.939 | 1.060 | 0.000 |
| | Risk × Vulnerable other | −0.89 | 0.66 | 0.181 | −0.411 | 2.181 | 0.003 |
| | Risk × Lack of privacy | −0.02 | 0.52 | 0.962 | −1.037 | 0.988 | 0.000 |
| | Risk × Friends | 0.43 | 0.45 | 0.345 | −1.317 | 0.461 | 0.001 |

Note. CI = confidence interval; LL = lower limit; UL = upper limit. All analyses were corrected for by age, gender, and education level. Higher scores on contact with household members or friends indicate worse contact.

### 3.2.2. Developmental Milestones

Results from the second multiple regression analysis show that youth who experienced a lack of privacy reported more difficulties in achieving personal milestones, with a small effect size (see Table 3). Contact with household members, living with a vulnerable other, contact with friends, neighbourhood risk, and the moderation effect of neighbourhood risk are not associated to the achievement of personal milestones. Youth who lived with a vulnerable other and youth who experienced a lack of privacy reported more difficulties achieving school and professional milestones. Additionally, neighbourhood risk moderated the association between living with a vulnerable other and achievement of school and professional milestones. Youth who lived in a low-risk neighbourhood reported similar difficulties in achieving school and professional milestones, whether they lived with or without a vulnerable other. Youth who lived in a moderate- or (extremely) high-risk neighbourhood reported more difficulties achieving school and professional milestones when they lived with a vulnerable other. For a graphical representation of the interaction effect, see Figure 2. All effect sizes are small. Contact with household members, contact with friends, neighbourhood risk, and the remaining interactions are not associated to the achievement of school and professional milestones. Youth who had a worse contact with friends reported more difficulties in achieving social milestones, with a small effect size. Contact with household members, living with a vulnerable other, a lack of privacy, neighbourhood risk, and the moderation effect of neighbourhood risk are not associated to the achievement of social milestones. In conclusion, the hypothesis on personal milestones was supported for a lack of privacy. The hypotheses on contact with household members, living with a vulnerable other, contact with friends, neighbourhood risk, and the moderating effect of neighbourhood risk were not supported. The hypotheses on school and professional milestones were supported for living with a vulnerable other, a lack of privacy, and the moderating effect of neighbourhood risk on living with a vulnerable other. The hypotheses were not supported for contact with friends, neighbourhood risk, and the moderating effect of neighbourhood risk on household (other than living with a vulnerable other) and friend factors. The hypothesis on social milestones was supported for contact with friends, but the other hypotheses were not supported.

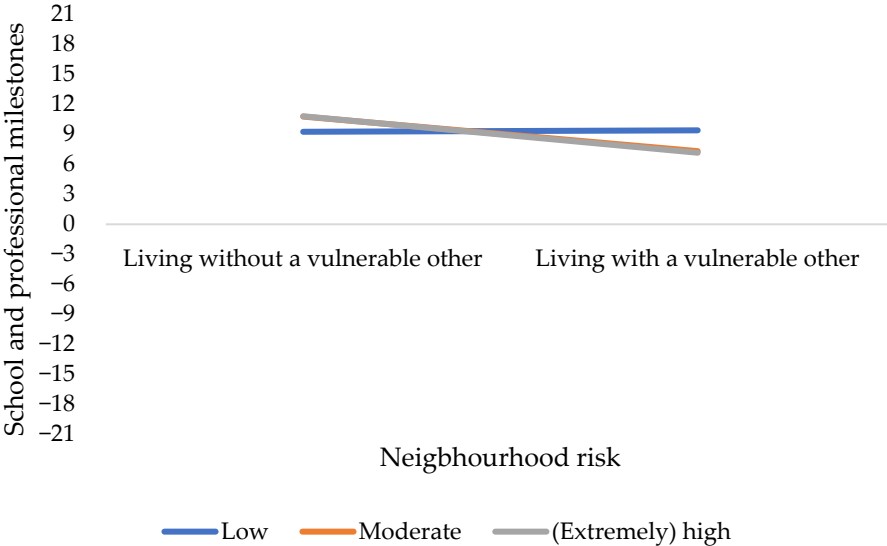

**Figure 2.** The association between living with or without a vulnerable other and school and professional milestones, moderated by neighbourhood risk.

## 4. Discussion

In the current study, we assessed two key elements of youth's functioning: youth's mental problems (i.e., internalizing and externalizing symptoms) and age-adequate functioning (i.e., youth-specific personal, school and professional, and social milestones). Specifically, it was examined how youth's layered social environments, in terms of their household, friends, and neighbourhood, were related to youth's mental problems and achievement of developmental milestones. As youth are characterized by large social role changes [3], the layered social environments were studied simultaneously to compare their associations to youth's functioning. We studied this during the COVID-19 crisis to obtain a more comprehensive understanding on how youth are functioning during crises.

Confirming our hypotheses, the current study indicated that youth experienced more internalizing symptoms when they perceived their contact with household members as worse during the lockdown compared to before. This is in accordance with previous studies that showed that the quality of the relationship with household members or support from them during a crisis is related to internalizing symptoms, such as depression [5,8]. For example, it is suggested, as this finding is replicated for different crises, to incorporate this knowledge in prevention strategies for mental health during a crisis. Contrary to our expectations, however, perceived contact with household members was not related to externalizing symptoms and personal, school and professional, and social milestones. This might be explained by contact with household members being measured as contact with household members during the lockdown compared to before the lockdown, while in many previous studies contact with household members was measured with reference to a longer timeframe (e.g., [9,39]). This suggests that a temporary worsened contact with household members is differentially associated with internalizing problems and other elements of youth's functioning, which could indicate that youth can deal well with temporary worsened contact with household members in most elements of their functioning. Another explanation might be our conservative approach using a more stringent *p*-value of 0.01. However, it is clear that—although in studies with a smaller number of variables, the association with externalizing symptoms and school and professional milestones was demonstrated in times of crisis [9,38]—it is not a substantial association.

As expected, youth who lived with someone they considered vulnerable, for example, because of an underlying condition, experienced more internalizing symptoms and reported more difficulties achieving school and professional milestones. The finding on internalizing symptoms is in accordance with previous studies that showed that during a crisis, poorer physical or mental health of a family member was related to youth inter-

nalizing symptoms [13–15]. The finding on difficulties achieving school and professional milestones is in line with a study that showed that children's school performance became poorer when their parent had a mental illness [41]. The current study, however, is the first to show this for age-adequate functioning during a crisis. It is important to also study this in other crises to understand if this holds true for other crises as well, and if yes, think about ways to prevent negative effects on youth. Contrary to our expectations, living with a vulnerable other was not related to externalizing symptoms and achieving personal and social milestones. Based on previous research that showed that in youth with a parent with a mental disorder, more parental monitoring was related to fewer externalizing symptoms [58], it is suggested that youth in the present study might have been less vulnerable for externalizing symptoms as there was potentially more parental monitoring during the COVID-19 crisis. For future research, examining how parental and school monitoring could work as a protective factor in times of crises may provide insight into how this could be helpful for youth's externalizing problems during times of crisis or heightened stress.

The data also indicate that youth who experienced a lack of privacy during the crisis reported more internalizing and externalizing symptoms and more difficulties achieving personal and school and professional milestones. These findings were as expected [16,18,42]. The present study, however, is the first that demonstrated these associations empirically. This indicates that this understudied factor deserves more attention in future research, also because it could shed new light on underlying mechanisms in the literature looking at different social constructs in relation to functioning. Furthermore, the finding that the lack of privacy is standing out as the factor in the present study that is most consistently related to youth's functioning, both in terms of mental problems and difficulty to achieve developmental milestones (except for social milestones), is slightly surprising. It is well known that youth are in a developmental period in their life in which they have to balance an emerging sense of self as a competent, autonomous individual on the one hand and feeling connections with significant others on the other hand, all for psychological well-being [59,60]. In the present study it seems, however, that needing privacy is more important than, for example, the experience of good contact with household members or friends on all elements of youth's functioning during the COVID-19 crisis. It might be that due to the social restrictions during the COVID-19 crisis, with youth and their household members spending more time at home, a lack of privacy was more problematic for youth. However, it is also imaginable that during other crises or stressful times, household members spend more forced time together, where the need for privacy remains important for youth. In future studies, it should be assessed whether this finding is COVID-19 crisis-related or also related to other crises or other situations with heightened stress.

According to our hypothesis, youth who perceived their contact with friends as worse reported more difficulties achieving social milestones. This is in line with a previous study that showed that youth who had a better relationship with friends experienced fewer difficulties achieving social milestones [40]. The present study adds to the existing knowledge that this is also true during the COVID-19 crisis. Whereas in other crises, physical social contact with friends might be affected negatively for different reasons than during the COVID-19 crisis, it might still affect youth in a similar way as in the COVID-19 crisis. Contrary to what was expected, however, perceived contact with friends was not related to mental problems and difficulties achieving personal and school and professional milestones. With a similar explanation as with contact with household members, this might be explained by contact with friends being measured with reference to a shorter timeframe compared to a longer timeframe (e.g., [40,43]). This could indicate that youth can deal well with temporary worsened relationship quality in most elements of their functioning.

Confirming our hypothesis, neighbourhood risk moderated the association between living with a vulnerable other and school and professional milestones. Specifically, youth who lived in a low-risk neighbourhood reported similar difficulties in achieving school and professional milestones, regardless of whether they lived with or without a vulnerable other. However, youth in moderate- or (extremely) high-risk neighbourhoods reported

more difficulties achieving school and professional milestones when living with a vulnerable other. This suggests that during the COVID-19 crisis, the combination of living with a vulnerable other in a riskier neighbourhood was particularly related to difficulties in achieving school and professional milestones. This finding confirms and extends a previous study indicating that youth exposed to cumulative effects, specifically during a crisis, have stronger reactions [61]. Similar results were found in a previous study that showed that youth from families with a lower socioeconomic status (SES) experienced greater losses on achievement outcomes when living in disadvantaged areas. This interaction was partly explained by the role of parents as moderators of contextual effects, as lower-SES parents may struggle to provide extra learning support or activities, thus being unable to counterbalance negative neighbourhood effects [62]. It might be that, in a similar way, youth in low-risk neighbourhoods have many resources that provide support in caring for a vulnerable other, while those in riskier neighbourhoods have less resources, and therefore less energy or time to spend on their school and professional milestones. This suggests that youth in riskier neighbourhoods who live with a vulnerable other could benefit from extra help, and it would seem sensible to pay attention to it in the development and execution of preventive care. Contrary to the expectations, neighbourhood risk was not directly related to mental problems, personal milestones, and social milestones. Additionally, no other significant interaction effects were found. One possible explanation is that our study had a large percentage of youth living in low- and moderate-risk neighbourhoods (±90%) and only a small percentage of youth who lived in a (extremely) high-risk neighbourhood (±10%). Even though this distribution is representative of Dutch society [48], the sample size of the (extremely) high-risk neighbourhood might have been too low to detect significant differences. Given that youth in (extremely) high-risk neighbourhoods face a stronger disadvantage compared to youth from low- and moderate-risk neighbourhoods [21], researchers should consider including more youth from (extremely) high-risk neighbourhoods in their research.

The impacts of crises and other extreme events are "sudden, inconceivable, damaging, sensitive, and unique" [63] (p. 205). Crises have in common that they often affect people's life in various ways [64]. People need to adapt very abruptly to the changing circumstances, which requires resilience [65]. How the threat is posed might differ between crises. As such, the impact of the COVID-19 crisis, which primarily poses a longer-term health threat, may not generalize to other types of crises such as earthquakes, which mainly pose an acute threat to physical safety. Additionally, where in the COVID-19 crisis people were confined to certain places, in other crises such as earthquakes, they might be displaced. In this last case, a lack of privacy might be better understood in relation to strangers instead of household members, as previously shown to be a problem for refugees in asylum centra [66]. Furthermore, the impact of the COVID-19 crisis might also depend on the area of the world. The current research took place in the Netherlands, with individualism as a dominant cultural paradigm. In more collectivistic cultures, the lack of privacy during the COVID-19 crisis might be experienced as less impactful, as the family in general is characterized by intergenerational interdependence [67]. Additionally, the Netherlands is a high-income country in the global north, for which results might not generalize to middle- or low-income countries in the global south. It would be important for researchers to do comparative research, for example, by building forth on the present study and a partly similar study that was performed in the global south [68]. Taken together, the present study suggests important factors to consider for youth from layered social environments during any crises and the COVID-19 crisis in particular, in relation to their mental problems and age-adequate functioning. This is especially important as youth mental health is a global health issue, which receives attention from both the United Nations [69] and World Health Organization [70] to improve it.

*Strengths and Limitations*

Some strengths and limitations should be kept in mind. One of the unique aspects of this study is the focus on age-adequate functioning in addition to the more often studied mental problems, which enabled us to examine different elements of youth's functioning during a crisis. Additionally, we studied layered social environments of youth simultaneously, thereby being able to compare each of the environments for different elements of youth's functioning. Furthermore, in designing the study, we collaborated with a youth panel. Based on the collaboration, we assessed youth's perceived lack of privacy during the crisis and showed that this is an important construct in assessing elements of youth's functioning in the COVID-19 crisis. In addition, we assessed the neighbourhood with objective, statistical data on multiple dimensions, which provides greater coverage of the construct and potentially knowledge on the concentration of risk factors. Together these strengths take the existing research a step further by paving the way for future research focusing more on youth's age-adequate functioning and thereby considering layered social environments simultaneously, both in times of heightened stress and relative quietness.

Despite these strengths, the current study also has limitations that need to be kept in mind when interpreting the results. Firstly, the assessment of youth's age-adequate functioning was conducted at a single timepoint, which makes causal interpretations impossible. Longitudinal research would be needed to assess youth's age-adequate functioning over time during crises and compared to the time pre- and post-crisis. This would shed light on the predictive nature of achieving developmental milestones for youth's future functioning. Another limitation is that a large part of the study, except for the neighbourhood risk index, relied on self-report. Additionally, perceived contact with household members, living with a vulnerable other, a lack of privacy, and perceived contact with friends were only measured with a single item, which increases the possibility of measurement inaccuracies. Future studies should aim to utilize more robust and comprehensive measures for a more reliable assessment. Additionally, it is worth noting that the sample predominantly consisted of girls. A previous meta-analysis has shown that female youth were more likely to experience a decline in mental health during the COVID-19 crisis [71]. To gain a comprehensive understanding, future studies should strive to include more boys and give attention to gender-based differences in times of crisis. Furthermore, all demonstrated effects were small, indicating that they only have limited practical or theoretical applications relative to the other effects in the study. Lastly, the research took place during the COVID-19 crisis, and even though it shares characteristics with other crises, it is unknown whether the findings can be generalized to other crises. It is recommended that future research explores elements of youth's functioning in various crises to better understand which associations are common to crises and which are more specific to a particular crisis.

## 5. Conclusions

In conclusion, the current research assessed two key elements of youth's functioning: youth's mental problems (internalizing and externalizing symptoms) and age-adequate functioning (i.e., personal, school and professional, and social milestones) and how these were associated to youth's layered social environments of their household, friends, and neighbourhood. Specifically, we assessed perceived contact with household members, living with a vulnerable other, experienced lack of privacy, perceived contact with friends, and neighbourhood risk. This was studied during the COVID-19 crisis to obtain a more comprehensive understanding on how youth are functioning during crises. In sum, from the associations of youth's functioning with their social environments, a lack of privacy and living with a vulnerable other provide the biggest burden on youth's functioning.

**Supplementary Materials:** The following supporting information can be downloaded at: https://www.mdpi.com/article/10.3390/adolescents3020025/s1. Neighbourhood risk index.

**Author Contributions:** Conceptualization, L.A.C.v.E.-B., H.D.S., P.T.v.d.H., M.A.G.v.A. and O.M.L.; methodology, L.A.C.v.E.-B., H.D.S and O.M.L.; formal analysis, L.A.C.v.E.-B. and O.M.L.; investiga-

tion, L.A.C.v.E.-B.; resources, H.D.S. and O.M.L.; data curation, L.A.C.v.E.-B. and O.M.L.; writing—original draft preparation, L.A.C.v.E.-B.; writing—review and editing, L.A.C.v.E.-B., H.D.S., P.T.v.d.H., M.A.G.v.A. and O.M.L.; visualization, L.A.C.v.E.-B.; supervision, H.D.S., P.T.v.d.H., M.A.G.v.A. and O.M.L.; project administration, L.A.C.v.E.-B.; funding acquisition, H.D.S., P.T.v.d.H. and O.M.L. All authors have read and agreed to the published version of the manuscript.

**Funding:** This research was funded by internal funding from the Faculty of Social and Behavioural Sciences of Utrecht University.

**Institutional Review Board Statement:** This study was conducted in accordance with the Declaration of Helsinki and approved by the Ethics Committee of the Faculty of Social and Behavioural Sciences of Utrecht University (protocol code: 20-413, date of approval: 1 October 2020).

**Informed Consent Statement:** Informed consent was obtained from all subjects involved in this study.

**Data Availability Statement:** The data presented in this study are available on request from the corresponding author. The data are not publicly available due to privacy and ethical restrictions.

**Acknowledgments:** The authors thank the youth who participated in this study, Linda van Tilburg (regional public health service), and Nathalie van den Bogaard-van der Meijden (school for Intermediate Vocational Education) for their support in facilitating the data collection. The authors thank Sander Thomaes for the help in the grant application and his general support in this project. We would also like to thank Emma Roza, Marije ten Den, and Marjolein van Cappellen for their contribution in the early stages of our research, and Karen Rienks for the help with further developing the neighbourhood risk index.

**Conflicts of Interest:** The authors declare no conflict of interest. The funders had no role in the design of the study; in the collection, analyses, or interpretation of data; in the writing of the manuscript; or in the decision to publish the results. There are no relevant financial or non-financial competing interests to report.

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
