# Peer review of "Youth’s Social Environments: Associations with Mental Problems and Achievement of Developmental Milestones in Times of Crises"

_adolescents, doi:10.3390/adolescents3020025_

Round 1
Reviewer 1 Report
Comments and Suggestions for Authors
Dear Authors,
Congratulations on your extensive work, concerning signifficance of youth’s social environments: associations with mental problems and achievement of developmental milestones in times of crises.
I suggest some minor revision:
Introduction
Lines 160-167: As such, we built on and extended previous work by 161 examining youth’s mental problems and age-adequate functioning to get a better under-162 standing of how youth are functioning during a crisis. Moreover, we used multi-dimen-163 sional data (e.g., income, crime, health) used from the national statistical office to index 164 neighbourhood risk. Findings could inform on social environmental influences, which 165 may in turn shed light on policies to alleviate negative consequences of current and future 166 crises.
Should be placed before study hypotheses, following the study aims.
M&M:
Procedures: Since some study participants were under 18 yrs of age, did authors obtain parents’ informed consent??
How about adding a participant flowchart, showing recrutiment to the study in details??
Line 232. Which version of SDQ was administered??
Results;
3.1. Correlation analysis in my opinion should be placed following the Main analyses.
Discussion:
The last part of discussion should be dvided into subsections: strenghts and limitations, future research implications, practical clinical implications, and then study conclusions.
Reviewer 2 Report
Comments and Suggestions for Authors
Comments for authors:
I reviewed the submission entitled "Youth’s social environments: associations with mental problems and achievement of developmental milestones in times of crises." This manuscript reports an analysis of how the social environment of youth’s during the COVID-19 pandemic impacted their behavioral health and developmental functioning. Overall, my impression of this manuscript was that it provides an intriguing contribution to the field of research surrounding youth development. I appreciate the authors’ efforts to thoroughly examine relevant factors that impact youth’s functioning during crises (i.e., contact with household members, lack of privacy, etc.). However, I have some concerns about the manuscript as submitted. I provide some minor comments worthy of attention below, which I hope will be useful to the authors and to the action editor. I appreciate having had the opportunity to review this work and wish the authors well in their research program.
General Comments
1. Please review the manuscript for minor errors and opportunities to improve clarity through grammar and sentence structure.
Introduction
2. Please consider restructuring the introductory section to improve the manuscript’s overall succinctness (e.g., lines 57 to 61).
Materials & Methods
2. Consider providing further description of the psychometric properties of the measures used within the study (e.g., the SDQ and DML).
Results
3. Please provide quantitative data within the description of findings (e.g., “with a small effect size” on lines 303 to 304; “All effect sizes were small” on line 315).
Discussion
4. While it may be uncertain regarding the generalizability of these findings to other crises (lines 479 to 480), please speak further to the potential application of this study’s findings to other potential crises.
Reviewer 3 Report
Comments and Suggestions for Authors
General comments
The authors ought to be commended for researching a topic of global relevance for understanding and promoting the well-being of youth. The paper is well-written. I have a few comments to help the authors strengthen their paper.
Introduction
The paper is well introduced. The limitations of the current literature are clearly identified and adequate justification is provided for the study. The research aim and hypotheses are well articulated.
Methods and materials
The description and justification of choice of specific methods are adequate.
Much of the content of section 2.1 seems to be the outcome of statistical analyses and should go to the results section. As a suggestion, this content could be merged with section 2.2 under a combined heading “Participants and procedure”.
Considering that the Developmental Milestones List (DML) is new and was developed specifically for this study, I was expecting to see some content on indicators of internal consistency and validity (e.g., content and face validity) in this section or somewhere in the prat of the Results for what this instrument measured. This would be useful to other researchers who might want to use the DML in their studies.
Results
Ideally, descriptive statistics associated with demographics should be put first to provide a good background/context within which to situate the study findings. Thus, Section 3.2 may have to come first. Content on section 2.1 could be brought here as I mentioned earlier.
For each of the relevant sections of the results, it would be good to indicate whether the associated hypothesis was supported or not. This would guide readers as they transition to your discussion section.
Discussion
The authors have adequately positioned their findings against the existing literature by discussing the implications of their findings for the issues they identified in the front end of the manuscript.
The authors may need to discuss the effect of the largely female sample on their findings. This is especially important considering that there are variations in how males and females experience and respond to the issues investigated in the study. If they consider this to be beyond the scope of this paper, they might want to recommend that future studies give attention to gender-based differences in the issues under investigation.
Considering that this study focuses on youth, it would strengthen the paper if the authors stress, in the concluding sections, the global relevance of their research since youth mental health is a global health issue. They can do this by mentioning the usefulness of their findings in relation to the global mental health agenda and SDG3.
With the majority of youth in the world originating from and living in the global south, which has the higher burden of poor mental health and characteristics that are very different from the study setting (the global north), I’d love to see a recommendation for future comparative studies on the topic of this research.
Good job and best wishes.
